

# Fecal shedding of SARS-CoV-2 in infants born to SARS-CoV-2 positive mothers: a pilot study

Dylan K.P. Blaufus[1], Karen M. Kalanetra[1], Rosa Pesavento[2], Pranav Garlapati[2], Brittany C. Baikie[1], Kara M. Kuhn-Riordon[2], Mark A. Underwood[2] and Diana H. Taft[3]

[1] Department of Food Science and Technology, University of California, Davis, CA, United States of America
[2] Department of Pediatrics, University of California, Davis, Sacramento, CA, United States of America
[3] Department of Food Science and Human Nutrition, University of Florida, Gainesville, FL, United States of America

Corresponding author
Diana H. Taft, dianataft@ufl.edu

## ABSTRACT

**Background.** Fecal shedding of SARS-CoV-2 occurs during infection, particularly in pediatric populations. The gut microbiota are associated with resistance to enteric pathogens. COVID-19 is associated with alterations to the gut microbiome. We hypothesized that the gut microbiome of infants born to SARS-CoV-2+ mothers differs between infants with and without fecal shedding of the virus.

**Methods.** We enrolled 10 infants born to SARS-CoV-2+ mothers. We used qPCR on fecal RNA to test for SARS-CoV-2 and 16S rRNA gene sequencing of the V4 region to assess the gut microbiome. Infant SARS-CoV-2 status from nasal swabs was abstracted from medical records.

**Results.** Of the 10 included infants, nine were tested for SARS-CoV-2 by nasal swab with 1 testing positive. Four infants, including the nasal swab positive infant, had at least one sample with detectable levels of SARS-CoV-2 fecal shedding. Detection of both SARS-CoV-2 genes in feces was associated with increased gut alpha diversity compared to no detection by a linear mixed effects model ($p < 0.001$). Detection of both SARS-CoV-2 genes was associated with increased levels Erysipelotrichaceae, Lactobacillaceae, and Ruminococceae by MaAsLin2.

**Conclusion.** Fecal shedding of SARS-CoV-2 occurs in infants who test negative on nasal swabs and is associated with differences in the gut microbiome.

## INTRODUCTION

Severe acute respiratory syndrome coronavirus 2 (SARS-CoV-2), the virus that causes Coronavirus Disease 2019 (COVID-19), first emerged in Wuhan, China in late 2019 (*Huang et al., 2020*). Since the emergence of SARS-CoV-2, it has become clear that this virus impacts more than just the respiratory system. For example, SARS-CoV-2 infection may include cardiovascular and gastrointestinal symptoms (*Lai et al., 2020*). SARS-CoV-2 infects cells by binding to the angiotensin-converting enzyme 2 receptor (ACE2). Levels

**How to cite this article** Blaufus DKP, Kalanetra KM, Pesavento R, Garlapati P, Baikie BC, Kuhn-Riordon KM, Underwood MA, Taft DH. 2024. Fecal shedding of SARS-CoV-2 in infants born to SARS-CoV-2 positive mothers: a pilot study. *PeerJ* 12:e17956
http://doi.org/10.7717/peerj.17956

of ACE2 depend in part on age, which is thought to be part of why children have less severe disease than adults (*Silva et al., 2023*). ACE2 is expressed in the gastrointestinal tract as well as the lungs, with ACE2 expression in the gastrointestinal tract higher in children than in adults (*Berni Canani et al., 2021*). Gastrointestinal symptoms in children infected with SARS-CoV-2 are common (*Al-Beltagi, AS & El-Sawaf, 2021*). Gut infection with SARS-CoV-2 results in fecal shedding of the virus (*Jones et al., 2020*), which is why wastewater tracking is useful to monitor community infection levels (*Hamouda et al., 2021*).

Early work has suggested that COVID-19 is also associated with the gut microbiome (*Zuo et al., 2021*). Severity of COVID-19 is associated with alterations in the gut microbiome, with severely ill patients having higher levels of opportunistic pathogens and lower levels of *Faecalibacterium prausnitzii* than controls in an adult patient population (*Tang et al., 2020*; *Zuo et al., 2020*). Children also experience dysbiotic gut microbiome changes with COVID-19, with a reduction of *Abiotrophia* in patients compared to controls (*Romani et al., 2022*). Infants hospitalized for COVID-19 had reduced levels of *Bifidobacterium* compared to healthy controls (*Gutiérrez-Díaz et al., 2023*). In one study, there was not an association between alpha or beta diversity of the infant stool microbiome and maternal history of SARS-CoV-2 during pregnancy, but infants born to mothers with a history of SARS-CoV-2 had higher levels of *Enterococcus* and lower levels of *Haemophilus parainfluenzae* (*Leftwich et al., 2023*). As this last study included samples from infants whose mothers were infected with SARS-CoV-2 during any trimester of pregnancy, it could not address the question of fecal shedding of SARS-CoV-2 in infants, as the majority of mothers cleared the infection prior to delivery. The gut microbiome has strong ties to maintaining the homeostasis of the immune system, including modulation of respiratory illnesses (*Budden et al., 2017*; *Hussain et al., 2021*). Communication between gut microbiota and the respiratory tract, known as the gut-lung axis, provides insight into the connection between respiratory illness and gastrointestinal symptoms (*Mindt & Di Giandomenico, 2022*). Because of the gut microbiome's role in immune defense and metabolism, the gut-lung axis is particularly relevant when there are disruptions of gastrointestinal microbes, as these disruptions can lead to increased risk of lung infection (*McAleer & Kolls, 2018*), including in infants (*Moroishi et al., 2022*).

Fecal shedding of SARS-CoV-2 may occur for days or months after the resolution of respiratory symptoms (*Natarajan et al., 2022*). Children exhibit prolonged fecal shedding of SARS-CoV-2, and testing of fecal samples for SARS-CoV-2 has been suggested to test for the clinical disease even with a negative nasal swab test (*Santos et al., 2020*). Vertical and intrapartum transmission of SARS-CoV-2 are possible but rare, although infant infection is typically tested for using an oropharyngeal swab or nasopharyngeal swab (*Egloff et al., 2020*; *Facchetti et al., 2020*; *Maeda et al., 2021*). Given that the gut microbiome is associated with SARS-CoV-2 infection and that young children may exhibit viral fecal shedding even with a negative nasopharyngeal swab, we hypothesized (1) that infants born to SARS-CoV-2 positive mothers may exhibit fecal shedding of SARS-CoV-2 even in the absence of a positive nasopharyngeal swab test and (2) that differences in fecal shedding of SARS-CoV-2 would be associated with differences in the gut microbiome of infants.

## MATERIALS & METHODS

### Recruitment and data collection

Infants were recruited from the University of California Davis medical center from May 2020 to August 2021. All infants born to mothers testing positive for SARS-CoV-2 at delivery were eligible for inclusion. We collected information on infant age at delivery, gender, the result of any clinical COVID-19 test, reason for infant hospitalization, and, if relevant, infant cause of death. All nasopharyngeal tests were the clinical test ordered by medical doctors and completed per hospital protocols, no additional nasopharyngeal tests were completed for this study. Results of nasopharyngeal tests were abstracted from the medical record. Written informed consent was obtained from one parent of each enrolled infant, this study was approved by the UC Davis IRB, #1585146.

### Sample Collection and processing

Stool samples were collected twice a week for up to eight weeks after infant enrollment in the study. If an infant was hospitalized, nurses completed collection of samples using the Zymo DNA/RNA shield buffer (ZR1100-50) aliquoted into two mL cryovials by rolling a flocked swab in the infant feces and placing the swab into the cryovial and shaking well. At infant hospital discharge, parents were provided with the option of additional collection kits to obtain the remaining samples using the same method. All samples were transferred from the hospital or home to the laboratory within 1 week, where samples were frozen at $-80\ °C$ until further use. Because the Zymo DNA/RNA shield buffer stabilizes nucleic acids at room temperature, all samples were stored at room temperature prior to transfer to the laboratory.

Viral RNA was extracted from the samples with the Zymo Quick-Viral RNA kit (R1034) following the kit's instructions and included an on-column DNase I treatment. RNA was eluted in 20 μL of nuclease-free water and stored at $-80\ °C$ until rRT-PCR assays were performed. rRT-PCR was carried out in triplicate with the CDC 2019-nCoV Real-Time RT-PCR Diagnostic Panel and AgPath-ID One-Step RT-PCR kit reagents. The panel includes three sets of primers and probes that target viral nucleocapsid gene loci - N1, N2 - and human RNase P gene loci as well as a positive control. Reaction conditions were one cycle of reverse transcription at $45\ °C$ 10 min, one cycle reverse transcription inactivation/ initial denaturation at $95\ °C$ for 10 min, 40 cycles of amplification and extension at $95\ °C$ for 15 s and $60\ °C$ for 45 s. Nuclease free water and RNA extracted from a SARS-CoV-2 negative adult human stool sample were used as negative controls. A sample was considered mixed if either N1 or N2 was detected in a sample, and positive if both were detected.

DNA extraction of stool samples was completed using the KingFisher Flex System (Thermo Fisher) with the ZymoBIOMICS 96 MagBead DNA Kit (D4308). The V4 region of the 16S rRNA gene from the resulting DNA samples was amplified and sequenced as described in *Heiss et al. (2021)*. Two negative extraction controls, consisting of an unused swab opened and placed into lysis buffer without swabbing a diaper sample were extracted for each plate run on the KingFisher. A positive control, consisting of the ZymoBIOMICS microbial community standard (D6300; Zymo Research) was also included on each KingFisher plate. For each PCR plate, two PCR negative controls consisting of water in

place of an extracted sample was included with the preparation. An additional PCR positive control consisting of the ZymoBIOMICS microbial community DNA standard (D6305, Zymo Research) was included on each PCR plate.

## Data analysis

Sequencing reads were analyzed using QIIME2-2020.2 (*Bolyen et al., 2019*). DADA2 was used to trim and truncate the data. After processing with DADA2 and assigning taxonomy with the QIIME2 prebuilt SILVA132 classifier, we exported the data and most subsequent analyses were completed using R version 4.2.3 (2023-03-15) (*R Core Team, 2023*). We used the results of the negative extraction and PCR controls in combination with decontam (*Davis et al., 2018*) to ensure a clean dataset. The prevalence method was used for decontam, with default settings (threshold =0.1). All samples were then rarefied to a depth of 6,437 reads before further analysis. Samples with less than 6,437 reads were excluded from additional analysis. Raw sequences are available from SRA accession number PRJNA1068637.

The vegan package (version 2.6.4) (*Oksanen et al., 2022*) was used to calculate alpha diversity by the Shannon Diversity Index. To account for the longitudinal nature of our data, we created a linear mixed effect model with a random intercept using the lme4 package (version 1.1.32) (*Bates et al., 2015*). The outcome variable of the model was alpha diversity, and stool SARS-CoV-2 detection was the primary predictor variable, with infant subject ID included in the model as a random effect.

Beta diversity was measured using the Bray-Curtis Distance and calculated using the vegan package in R. The resulting matrix was visualized using Non-Metric Multidimensional Scaling, with the number of dimensions needed for the graph chosen using a Scree plot. Samples were then stratified by time point, with only samples collected from end of week 2 to the start of week 5 eligible for inclusion. This time range was selected because it maximized the number of positive samples available for inclusion and the number of infants with a sample available for inclusion. Only one sample per infant was included, selected at random if an infant had more than one sample from the time range available. One infant was excluded from the beta-diversity analysis due to the lack of available samples during the time range selected for analysis. The adonis function in vegan was used to test for differences in beta-diversity by fecal SARS-CoV-2 shedding status and sample collection time point.

Differential abundance of taxa in samples with and without fecal shedding of SARS-CoV-2 was assessed using MaAsLin2 in R version 4.2.3 (*R Core Team, 2023*) using a mixed-effects model with fecal sample SARS-CoV-2 result and sample collection time as fixed effects and infant ID as a random effect to account for repeated measures. Default MaAsLiN2 settings were used. Model was run on the unrarefied dataset, at the family level.

## RESULTS

### Enrolled infants and fecal shedding of SARS-CoV-2

We recruited 10 infants for inclusion in our study. Gestational age at birth of enrolled infants ranged from 25 to 39 weeks. Table 1 shows the collected demographic information
**Table 1  Included infant demographics.**

| Number of infants | 10 |
|---|---|
| Infant Sex | 3 Female<br>5 Male<br>2 Unreported |
| Gestational Age | 37 Weeks (range 25 Weeks–39.14 Weeks) |
| Reasons for Hospitalization | Prematurity, Sepsis, Hyperglycemia, Respiratory Disease, Congenital Syphilis, Cyanotic Heart Disease |
| Median Samples Collected per Infant | 5 samples (range 1 –16 samples) |

**Figure 1  Illustration of infant age at sample collection, including infant hospitalization status, detection of SARS-CoV-2 in the fecal sample, and inclusion in the beta-diversity analysis.**

on the enrolled infants. Of the 10 infants, one infant died from sepsis, while the other infants survived until discharge from the NICU. Of the 10 infants, nine were tested for SARS-CoV-2 by nasal swab with one infant testing positive. In contrast, four of the 10 infants had detectable levels of at least one of the two SARS-CoV-2 genes at at least one time point. One of these four infants was the infant that tested positive *via* nasal swab, one was the infant not tested by nasal swab, and the other two infants had a negative nasal swab for SARS-CoV-2. Of the 61 fecal samples collected, six were positive by qPCR for both SARS-CoV-2 genes, with five samples being mixed with one positive and one negative SARS-CoV-2 gene. Figure 1 illustrates the timeline of samples collected for this study. The majority of infants were hospitalized, meaning that staff collected the majority of samples. As most of the positive samples were collected between the end of week 2 and the start of week 5, we randomly selected one sample per control infant to include in the beta-diversity analysis.

## Control sample results

Read depth in the negative control samples ranged from 83 to 2,346. Before decontam, only 0.8947% of ASVs were present in the negative controls. Decontam did not identify any ASVs as likely contaminants.

Read depth in the positive control samples ranged were 22,807 to 73,439. The Zymo Mock Communities (both extraction control and DNA control) were expected to contain by 16S rRNA gene sequencing the following: *Listeria monocytogenes* –14.1%, *Pseudomonas aeruginosa* –4.2%, *Bacillus subtilis* –17.4%, *Escherichia coli* –10.1%, *Salmonella enterica* –10.4%, *Lactobacillus fermentum* –18.4%, *Enterococcus faecalis* –9.9%, and *Staphylococcus aureus* –15.5%. The positive control samples at the genus level were found to have 22% *Listeria,* 5.8% *Pseudomonas,* 20% *Bacillus,* 7.8% *Escherichia-Shigella*, 9.6% unknown genus of family Enterobacteriaceae (*Salmonella* is in family Enterobacteriaceae), 14% *Enterococcus,* 20% *Staphylococcus*, and <1% a mixture of a large number of taxa detected at very low levels. Genus *Lactobacillus* was completely missing, suggesting that results relevant to *Lactobacillus* should be interpreted with care. As the mock community was correct at the family level other than the absence of Lactobacillaceae, we chose to report on the family level.

## Gut microbiome and SARS-CoV-2

Read depth in samples ranged from 6,437 to 136,924, with a read depth of 6,437 chosen for rarefaction for all subsequent microbiome data analyses except MaAsLin2. Figure 2 shows the relative abundance of the microbiota within the fecal samples. The most abundant taxa at the family level were Bifidobacteriaceae, Clostridiaceae, Enterobacteriaceae, Erysipelotrichaceae, Lachnospiraceae, Peptostreptococcaceae, Staphylococcaceae, Streptococcaceae, and Veillonellaceae.

The Shannon Diversity Index was used to measure alpha diversity in each sample. The median alpha diversity of samples negative for SARS-CoV-2 was 1.480, with a range of 0.326−2.619. For positive samples, the median alpha diversity was 3.068 with a range of 1.713−3.427. For mixed samples (meaning only one of the two SARS-CoV-2 qPCR targets amplified), the median alpha diversity was 1.307 with a range of 1.054−3.031. Figure 3 shows Shannon diversity by SARS-CoV-2 fecal shedding and by infant age. Linear mixed effect modelling found that fecal shedding of SARS-CoV-2 was associated with increased alpha diversity ($p < 0.01$) even after for adjusting for infant age at sample collection (Table 2).

The NMDS plots showed no visible separation of samples by fecal shedding status in any dimension (Fig. 4). As PERMANOVA does not have a longitudinal implementation, we selected a single time range to maximize the number of positive samples while including only one sample per infant. This time range meant only nine of the 10 infants had a sample available for inclusion, and we detected no significant difference in beta-diversity by SARS-CoV-2 fecal shedding status ($p > 0.05$), consistent with the lack of separation observed in the NMDS plot.

MaAsLin2 identified a number of differences related to sample time point and to fecal shedding. Positive fecal shedding was associated with increased relative abundance of
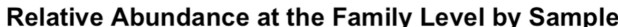

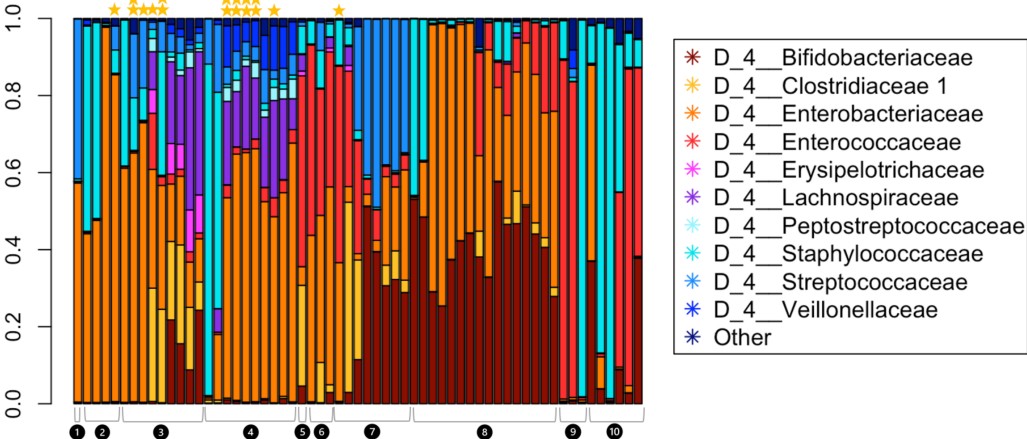

**Figure 2** **Relative abundance at the family level of bacterial taxa detected in the infant fecal samples.** Each bar represents a single sample, and samples are grouped and numbered by infant and ordered longitudinally, with the first sample collected for each infant leftmost. Two stars above a bar indicate a sample with detectable SARS-CoV-2. A single star indicates one of the two genes tested were present. No star indicates a negative sample.

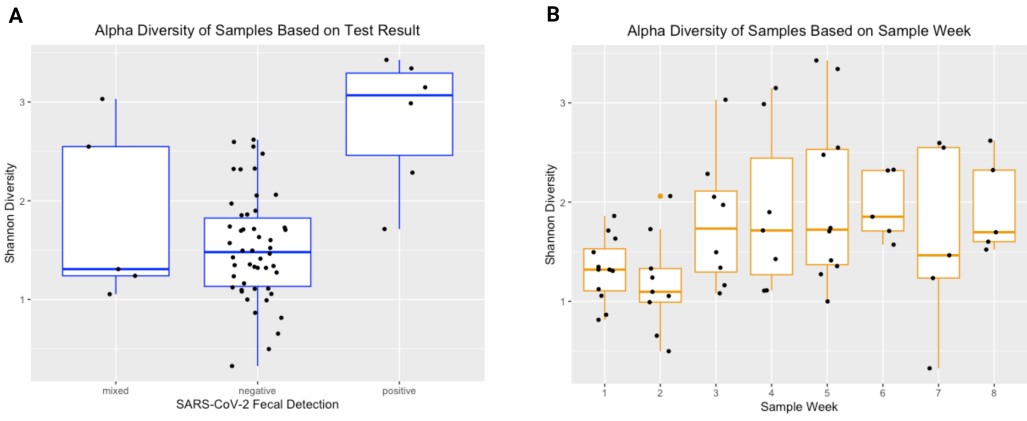

**Figure 3** **Alpha diversity of stool samples.** Alpha diversity of stool samples by SARS-CoV-2 fecal shedding status (A) and by infant age at sample collection (B).

Erysipelotrichaceae, Lactobacillaceae, and Ruminococceae. Increasing sample collection week was associated with decreased Staphylococcaceae and increased Enterococcaceae and Micrococcaceae (Table 3).

## DISCUSSION

The detection of SARS-CoV-2 in fecal samples from infants who tested negative on a nasopharyngeal swab in this study is consistent with prior reports in the literature, where persistent fecal shedding in children with a prior positive nasopharyngeal swab has been

**Table 2   Linear mixed effects model of alpha diversity by SARS-CoV-2 fecal shedding and time.**

|  | Estimate | Standard error | *t*-value |
|---|---|---|---|
| (Intercept)[*] | 1.19 | 0.169 | 7.02 |
| Feces positive for COVID vs negative[*] | 0.877 | 0.286 | 3.07 |
| Feces mixed for COVID vs negative | 0.166 | 0.218 | 0.761 |
| Time since study enrollment[*] | 0.0517 | 0.0127 | 4.08 |

**Notes.**
[*]Significant in model at $p < 0.01$.
[N.S.]Results have no superscript.

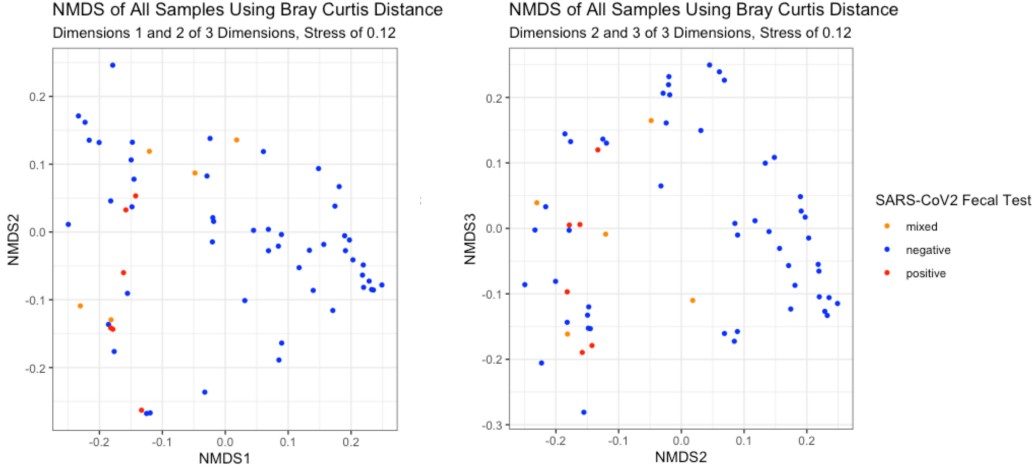

**Figure 4   Beta-diversity of infant stool samples.** NMDS ordinations based on Bray-Curtis distance. There was no visible separation by SARS-CoV-2 fecal shedding status.

**Table 3   Taxa at the family level identified by MaAsLin2 significantly associated with sample time point and detectable SARS-CoV2.**

| Taxa | Variable | Coefficient | Standard error |
|---|---|---|---|
| Erysipelotrichaceae[*] | Feces positive for COVID vs negative | 5.40 | 1.00 |
| Lactobacillaceae[*] | Feces positive for COVID vs negative | 4.24 | 1.02 |
| Ruminococceae[*] | Feces positive for COVID vs negative | 5.90 | 1.14 |
| Staphylococcaceae[*] | Time since study enrollment | −1.28 | 0.272 |
| Enterococcaceae[**] | Time since study enrollment | 1.07 | 0.314 |
| Micrococcaceae[**] | Time since study enrollment | 1.05 | 0.304 |

**Notes.**
[*]Significant with q-value $< 0.01$.
[**]Significant with a q-value $= 0.02$.

reported. Our detection of SARS-CoV-2 in infants with a negative nasopharyngeal swab suggests further exploration into the use of feces to detect SARS-CoV-2 infection in young infants is warranted, especially in light of the differences in ACE2 expression observed between children and adults (*Berni Canani et al., 2021*). This result stands in contrast with other work that did not find SARS-CoV-2 in the urine or feces of infants born to mothers who had COVID-19 during pregnancy (*Boateng et al., 2022*). However, as only six infants in that study of 42 infants were born to mothers who had COVID-19 at time of delivery and because the study was limited to term infants (*Boateng et al., 2022*), our results may suggest infection at delivery or preterm status impact the likelihood of detecting SARS-CoV-2 in infant feces. As there is at least one case report of a late-preterm infant developing SARS-CoV-2 after delivery by an infected mother despite isolation of the infant from the mother (*Kokoreva & Kazartseva, 2022*), there likely is a difference in infection risk with active infection at time of delivery or preterm status compared to term infants with mothers who had COVID-19 at some point during pregnancy. Fecal shedding is known to occur in infants infected with SARS-CoV-2, including in breastfed infants (*Holm-Jacobsen et al., 2021*; *Pace et al., 2024*). Our study suggests that infection at time of delivery may be associated with fecal shedding in the infant, but more work is needed to know if this might represent infectious virus or just viral remnants.

Alpha diversity is a measure of the total number of species present in a sample and their distribution. Prior work has demonstrated connections between alpha diversity and infectious illness in several contexts. For example, *Romani et al. (2022)* found that alpha diversity in the gut microbiome was lower in COVID-19 patients with mild to moderate symptoms compared to SARS-CoV-2 infected asymptomatic individuals. Other work on critically ill infants, conducted by Ojima et al., found that gut dysbiosis as measured by a reduction in alpha diversity was linked to an increase in severity of a variety of diseases including sepsis and acute respiratory illness. In a study conducted by *Moroishi et al. (2022)*, increased alpha diversity of the gut microbiome at six weeks of age was associated with increased risk of respiratory infection during the first year of life. A study that included comparisons of fecal shedding and non-shedding of influenza in adults found a reduction in alpha diversity in shedders. In the case of COVID-19, severe disease is associated with reduced alpha diversity. In addition, there is concern that the alterations to the gut microbiome of pregnant women could lead to differences in the gut microbes passed on to infants. In our study, fecal shedding of SARS-CoV-2 in infants was associated with greater gut microbiome alpha diversity, consistent with other respiratory illness in infants and the impact on the gut microbiome, but inconsistent with the adult findings relevant to were born to mothers with SARS-CoV-2 at the time of delivery. As a result, all mothers may have experienced a reduced alpha diversity at the time of delivery, and those infants who exhibited fecal shedding were the infants where maternal-infant transfer of microbes was most effective. There was no statistically significant difference in alpha diversity between the negative SARS-CoV-2 fecal samples and the samples with a mixed SARS-CoV-2 result. This could be because the mixed samples were truly negative for SARS-CoV-2 shedding, or it could mean that infants enrolled later in the study were infected with SARS-CoV-2 variants which were less efficiently detected by our qPCR and were truly positive. If the

latter, this would suggest that later variants had less association between gut microbiome and fecal shedding. Another possibility is that different variants exhibit different behavior in regards to fecal shedding or the gut microbiome, as there are known differences in fecal shedding rates between variants (*Prasek et al., 2023*), this possibility cannot be ruled out. Unfortunately, we do not have information on the variants detected by shedding, besides what can be guessed from the primary variants in circulation at time of enrollment. Our results in alpha diversity increasing with age are also consistent with what is reported in the literature.

Beta diversity is the measure of overlap in taxonomic composition in populations. In a past study conducted by *Al Khatib et al. (2021)*, beta diversity was not found to be linked with viral fecal shedding of influenza. However, studies to date have demonstrated that SARS-CoV-2 infection was correlated with beta diversity (*Romani et al., 2022*; *Wu et al., 2021*). Our research found no statistical significance in beta diversity by SARS-CoV-2 fecal shedding in infants. The difference in our findings may again be because of our study population –all infants were born to SARS-CoV-2 infected mothers, which likely results in differences in the microbes transmitted from mothers to infants. Alternatively, as all infants were asymptomatic for COVID-19, even infants who exhibited fecal shedding may not have been ill enough to experience substantial differences in the gut microbiome.

We also identified three families of microbes differentially abundant in infants who were positive for fecal shedding of SARS-CoV-2: Erysipelotrichaceae, Lactobacillaceae, and Ruminococceae. This is in contrast to results in children, where individuals with COVID-19 had reduced levels of *Ruminococcus* (*Romani et al., 2022*). However, the Romani study focused on the genus level and the children in the Romani study were older, with a median age of 6.5 years (*Romani et al., 2022*), which may mean other members of the Ruminococceae were impacted in our study and likely represents substantial differences in the microbiome from age alone. Further studies in infants are needed to understand infant specific differential abundance of microbes with fecal shedding of SARS-CoV-2.

The main limitation to this study was the small sample size. Efforts were made to recruit a larger population, but having mothers test positive for SARS-CoV-2 at time of delivery limited our ability to obtain consent for infant participation and we fortunately did not see infants hospitalized in the NICU for SARS-CoV-2 infection during the recruitment period. The small sample size also limited our ability to determine impacts of other potential confounding factors such as delivery type, feeding type, and duration of hospital stay. This means that our results may be confounded by differences in these factors. Unfortunately, we could not adjust for these factors without overloading the model. Furthermore, only one infant was not hospitalized during the study, making it difficult to adjust for the effect of hospitalization. Despite our small sample size, we were still able to detect microbiome differences associated with SARS-CoV-2 fecal shedding and to suggest that infant fecal shedding of SARS-CoV-2 may occur even in the absence of a positive nasopharyngeal swab test. An additional limitation is our failure to detect *Lactobacillus* in the Zymo positive controls. This means that our finding of an association between Lactobacillaceae and SARS-CoV-2 shedding must be interpreted with care, as there may have been more Lactobacillaceae in the non-shedding infants that went undetected.

## CONCLUSIONS

Fecal shedding of SARS-CoV-2 was associated with differences in infant gut microbiome alpha diversity and relative abundance of Erysipelotrichaceae, Lactobaceillaceae, and Ruminococceae. We did not detect any differences in beta-diversity associated with fecal shedding of SARS-CoV-2. Fecal shedding can occur even in infants who test negative for COVID19 on a nasal swab. Further research is needed to understand the connection between gut microbiome and fecal shedding of SARS-CoV-2 in newborns, but our findings suggest there is a connection between infant gut microbiome and fecal shedding.

## ACKNOWLEDGEMENTS

We are grateful to Dr. David A. Mills for the use of his laboratory and equipment to complete this project.

### Funding

This project was funded by a UC Davis COVID19 Research Accelerator Funding Track (CRAFT) grant. The funders had no role in study design, data collection and analysis, decision to publish, or preparation of the manuscript.

### Grant Disclosures

The following grant information was disclosed by the authors:
A UC Davis COVID19 Research Accelerator Funding Track (CRAFT).

### Competing Interests

The authors declare there are no competing interests.

### Author Contributions

- Dylan K.P. Blaufus performed the experiments, analyzed the data, prepared figures and/or tables, authored or reviewed drafts of the article, and approved the final draft.
- Karen M. Kalanetra conceived and designed the experiments, performed the experiments, analyzed the data, authored or reviewed drafts of the article, and approved the final draft.
- Rosa Pesavento performed the experiments, authored or reviewed drafts of the article, and approved the final draft.
- Pranav Garlapati performed the experiments, authored or reviewed drafts of the article, and approved the final draft.
- Brittany C. Baikie performed the experiments, authored or reviewed drafts of the article, and approved the final draft.
- Kara M. Kuhn-Riordon performed the experiments, authored or reviewed drafts of the article, and approved the final draft.
- Mark A. Underwood conceived and designed the experiments, performed the experiments, authored or reviewed drafts of the article, and approved the final draft.

- Diana H. Taft conceived and designed the experiments, performed the experiments, analyzed the data, prepared figures and/or tables, authored or reviewed drafts of the article, and approved the final draft.

## Human Ethics

The following information was supplied relating to ethical approvals (*i.e.,* approving body and any reference numbers):

This study was approved by the UC Davis IRB, #1585146.

## Data Availability

The sequencing data are available at NCBI SRA: PRJNA1068637.

The raw data on infant characteristics and fecal shedding are available in the Supplementary File.

## Supplemental Information

Supplemental information for this article can be found online at http://dx.doi.org/10.7717/peerj.17956#supplemental-information.

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
