# Peer review of "Fecal shedding of SARS-CoV-2 in infants born to SARS-CoV-2 positive mothers: a pilot study"

_PeerJ, doi:10.7717/peerj.17956_

## Round 0.1 · original submission · Major Revisions

Dear authors, please refer to the reviewers' comments for details. Make sure to address all the comments point by point and you may need to improve your discussion, particularly including more publications relevant to SARS-CoV-2 fecal shedding in infants.

Reviewer 1 ·

Basic reporting

- Please be consistent in the use of SARS-CoV-2 (often SARS-CoV2 is used).
- Line 77: missing a period at the end of sentence.
- Ensure that references related to SARS-CoV-2 fecal shedding in infants are included and up to date (e.g., PMIDs: 34772377, 34718351, and 38585272)

Experimental design

• Overall, as the authors acknowledge there is limited power with a sample size of 10 infants (and uneven longitudinal sampling).
• Opening sentence of abstract relates to GI symptoms, yet these do not appear to have been assessed.
• Methods are not described sufficiently.
• More details could be given for the viral RNA extraction methodology, e.g., were extracts DNAse treated? There are also no details given for how the detection/quantification of SARS-CoV-2 RNA was performed. This last piece is critical to understanding and interpreting the results.
• As the authors note, the inability to detect lactobacillus in the mock communities is odd. But there is no description in the methods on how taxonomy classification was performed or on the number of positive controls extracted (or negative controls). Is it possible that the database used was missing Lactobacillus (or Limosilactobacillus)? Or did the classifier fail to classify down to the genus level and only Lactobacillaceae was classified? What was the actual mock community that was extracted (i.e., catalog number)? This information should also be included.
• It appears that the repeated measures of samples were not accounted for in statistical models.

Validity of the findings

- The PeerJ data availability statement includes the accession for the microbiome sequence data. However this should be included within the manuscript as well.
- Metadata are included as a supplemental Rdata file. It may be more convenient to a larger swathe of readers to include these data in a tab-delimited format.

·

Basic reporting

The research is relevant as it addresses a gap in understanding the vertical transmission of SARS-CoV-2 and its effects on newborns. It provides insights into the diagnostic challenges posed by the virus, particularly in vulnerable populations like infants. The topic is quite original, as it looks into fecal shedding of SARS-CoV-2 in infants and its association with the gut microbiome.
The introduction provides a comprehensive overview of the current understanding of SARS-CoV-2 and its impacts beyond the respiratory system, particularly focusing on its effects on the gastrointestinal tract and the gut microbiome. The language is precise, and technical terms are used appropriately. However, the complexity and density of information could be streamlined for better readability. Breaking down long sentences and ensuring a smooth flow of ideas can make it more clear to the reader. For example:
"Severe acute respiratory syndrome coronavirus 2 (SARS-CoV-2), the virus that causes Coronavirus Disease 2019 (COVID-19), first emerged in Wuhan, China in late 2019."
There is a thorough review of existing literature. The hypotheses are clearly stated.

Experimental design

The methods section of this study is detailed and well-structured, providing a clear pathway for replication. The conclusions drawn in the discussion section are well-aligned with the evidence presented. The detection of SARS-CoV-2 in fecal samples from infants, despite negative nasopharyngeal swabs, is consistently supported by both the study's data and prior literature. This validates the conclusion that fecal testing may be a valuable diagnostic tool.

Validity of the findings

The discussion section of the article provides a thorough and insightful analysis of the study's findings, linking them effectively to existing literature and acknowledging limitations. The interpretation of the findings could be expanded. For instance, the discussion on why infants with fecal shedding exhibited greater gut microbiome alpha diversity than their non-shedding counterparts, despite the general trend of reduced diversity in severe COVID-19 cases, could be elaborated further to explore underlying mechanisms or hypotheses.

Additional comments

Minor Formatting Issues:
Ensure consistent formatting for terms like "SARS-CoV2" (should be "SARS-CoV-2").
Check for any typographical errors, such as "COVID19" which should be "COVID-19".

Reviewer 3 ·

Basic reporting

The authors present af study where the gut microbiome of infants born by SARS-CoV-2 positive mothers were analyzed. Specifically, the authors compared the microbiome of SARS-CoV-2 positive infant fecal samples to those that were negative. The research goal is interesting, however, the manuscript suffers from several limitations. A major drawback is the study design where the authors rely on longitudinal data from infants where changes in gut microbiota composition can severely affect their data and this in combination with the small number and heterogenous nature (different number of fecal samples) of samples, makes valid conclusions difficult. Furthermore, the methods section is very poorly described and several critical points and information are not given. Please see specific methods below.

Experimental design

Methods
Note that the references in this section are not numbered and therefore not included in the reference list.
Do you have information on the age of the children at the time point of fecal sample collection? Immediately after birth there is a rapid and dramatic change in the gut microbiome composition. Could this have affected your results? E.g. how old were the fecal+ infants compared to the fecal- infants? Were some of the samples meconium?
You have a very broad range in gestational age at birth. This is also known to affect the infant gut microbiome establishment. How did this affect your results?
Did the mothers or the infants receive antibiotics at any time point? Again this is important information when looking at microbiome data.
What about breastfeeding vs formula. Did any of the infants receive formula at any timepoint? This can affect the gut microbiome.
What about birth mode which is also an important impact factor on gut microbiome establishment in infants?
Were all infants hospitalized in the NICU and for how long were they hospitalized? The hospital environment can affect gut microbiome composition. Furthermore, were there any difference in storage time or temperature between hospital and home samples? Do you see any difference in microbiome composition between hospital collected samples and home collected ones?
How were the samples collected? In diapers? Who collected the samples? The mothers or staff? Were special measures taken to prevent cross contamination of virus from the mother to the fecal samples?
How was the test for SARS-CoV 2 performed? What were the used cut-off levels (test sensitivity etc.), and were the same experimental procedures used for both nasopharyngeal and fecal testing? More details are necessary here.
The authors state that they use negative extraction and PCR controls as negative controls for the Decontam procedure. These negative controls should be described in more detail: Does the negative extraction consist of Zymo DNA/RNA shield buffer without feces or is it lysis buffer that was run alone? What do the PCR controls refer to?
The authors state they utilized the decontam package together with the negative control. I therefore presume that the authors utilized the prevalence method for contaminant identification. Please state in the methods which contaminant identification method, and which threshold were used (default = 0.1).
What was the reasoning behind focusing on the family taxonomic level when looking at bacterial composition with barplot and MaAsLin2? The authors would have obtained a higher resolution by focusing on the genus level, which should have been possible to do when using 16S rRNA gene sequencing targeting the V4 region.
I don´t understand the sentence in lines 115-116.

Validity of the findings

Results
It is difficult to follow the study design and timeline. I would recommend including a timeline overview figure for instance showing for each infant 1) nasal and fecal SARS-CoV-2 status, 2) time from birth to first fecal sampling, 3) number of fecal samples (which are from the hospital and which are from home), 4) mark which samples are included in the PERMANOVA.
In line 157 Figure 4 should be changed to Figure 3.

Discussion
In lines 194-197: Do you have information on the variants found in your samples? Have other studies looked at fecal presence of SARS-CoV-2 and variant types?
In lines 201-203: Note that you only analyze data at family level whereas the other studies referred to analyze at genus level. You cannot compare.

---

## Round 0.2 · Minor Revisions

Dear authors, although I am only asking for minor revisions, I do urge you to proofread and actively correct the language in your manuscript. This includes checking that ALL the values that have been correctly output/registered, any figures have enough quality/ resolution.

Reviewer 1 ·

Basic reporting

Most of my comments and feedback have been sufficiently addressed. However, I have a couple of additional minor comments.
1. I did not see a csv file as indicated by the authors. The .Rdata file (and consent form for review) was the only supplemental file included.
2. The abstract text could be improved for grammar.
-The second sentence in the abstract include a semi colon. I would suggest breaking this into two full sentences.
-There is a strange line break on lines 42-43 that might just be a formatting issue.
-Check that formatting of references in consistent, e.g., line 129 includes a superscript reference
-Extra period on line 143
-Lines 173-179 - I do not believe that these are the correct values in the reference mock community and should be updated. The relative abundance values of Zymo D6300 listed are the % of the genomic DNA. The values should instead be the "16S only" theoretical %, given in Table 1 of the product's protocol. It is reassuring that the relative abundance of the taxa the authors report identifying in their extracted mock communities are relatively close to these reference values. I would also suggest omitting referencing the non-bacterial microbes from this discussion for brevity and lack of relevance to the method of detection.

Experimental design

no comment

Validity of the findings

no comment

---

## Round 0.3 · accepted · Accept

Dear authors. I am now accepting your work for publication in peerj. There are still minor revisions that can easily happen in the proofreading/production stage such as changing "u" by the correct microliter symbol, or the missing celsius symbols "º" around line 113, etc. Thank you for your submission.